# Impact of Providing Peer Support on Medical Students’ Empathy, Self-Efficacy, and Mental Health Stigma

**DOI:** 10.3390/ijerph19095135

**Published:** 2022-04-23

**Authors:** Matthew P. Abrams, Joshua Salzman, Andrea Espina Rey, Katherine Daly

**Affiliations:** 1College of Medicine, University of Central Florida, Orlando, FL 32827, USA; salzmanjo@knights.ucf.edu; 2Focused Inquiry & Research Experience Module Department, College of Medicine, University of Central Florida, Orlando, FL 32827, USA; andrea.espinarey@ucf.edu; 3Department of Clinical Sciences and Student Affairs, College of Medicine, University of Central Florida, Orlando, FL 32827, USA; katherine.daly@ucf.edu

**Keywords:** peer support, medical education, empathy, mental health, stigma

## Abstract

Background: Peer-support programs in medical school can buffer feelings of inadequacy, anxiety, social isolation, and burnout, drawing upon the benefits of near-peer-support resources. This study examined the effects of providing support to students in a medical school peer-support program. Methods: Using a pre-post, quasi-experimental study design, the investigators surveyed medical students who were peer supporters in their second through fourth years of medical school with four measures assessing (1) empathy, (2) self-efficacy, (3) mental health stigma, and (4) likelihood to assist peers with mental health problems to examine if serving as a volunteer peer supporter had any effect. Participants included 38 medical students that were actively enrolled peer supporters during the 2020–2021 year at a United States allopathic medical school. Results: Medical students who participated as peer supporters were found to have higher ratings of empathy scores (Z = −1.964, *p* = 0.050, r = 0.34) and self-efficacy scores (Z = −2.060, *p* = 0.039, r = 0.35) after participation in the program. No significant changes were noted for mental health stigma or likelihood to assist peers with mental health problems. Discussion: Peer-support programs present a low-cost, sustainable modality to promote wellbeing in medical students. There is a growing body of literature documenting the benefits of peer-support services. This brief, novel study examined the effects of providing peer support on the peer supporters and found higher self-reported ratings of empathy and self-efficacy after participation. These findings underscore peer-support programs as a valuable wellness resource not only for medical students who use the services but for those who provide them as well.

## 1. Introduction

Medical school is recognized as a difficult and stressful time for students [1,2,3]. Besides the stressors intrinsic to learning a fast-paced, rigorous medical curriculum, many students encounter a culture of competition at their institutions [4]. These factors contribute to imposter syndrome experienced by many students [5].

Further, even for academically adept students, these stressors and social interactions weigh heavily on students [6]. Many medical students report increased symptoms of depression [7], symptoms of anxiety [8], burnout [9], substance misuse [10], and suicidal ideation [11] compared to age-matched peers in the general population. For example, in a recent meta-analysis, the overall prevalence of depressive symptoms among medical students was 27.2%, and the overall prevalence of suicidal ideation was 11.1% [11]. Among medical students who screened positive for depression, only 15.7% sought psychiatric treatment [11]. Consistent with low rates of mental healthcare usage, other studies have found that mental health stigma may make some students reluctant to seek care [12] or disclose their mental health status [13]. These trends continue beyond medical school, as rates of physicians with depression are also higher than their age-matched peers [14]. Barriers to help seeking that have been identified in previous studies include concerns about confidentiality, time, cost, perceived stigma, potential repercussions, and fear of unwanted interventions [15,16]. These high rates of mental health symptoms underscore the need for effective prevention programs. Although medical students indicate social support and shared experiences as important resources to promote positive mental health, medical schools must implement effective wellness strategies within the learning environment to properly support their students [8]. 

Promoting the wellbeing of medical students and the physician workforce is important for patient care [17]; therefore, medical schools across the nation are attempting to reduce the high burden of mental health symptoms [18]. Much of the research highlighting mental health declines in medical school occurred prior to the COVID-19 pandemic, which has only further exacerbated these concerns by placing greater demands on health professionals and increasing social isolation. A positive trend since the pandemic has been the increase in distance learning [19] and telehealth, making support resources more easily available. Some institutions have implemented wellness interventions such as “resilience days” to fortify learners against burnout [20]. Recently, there has been an increase in student-led initiatives, such as student-led mental health workshops [21]. 

Peer-support programs—a newly employed wellness intervention in medical education—can be used to buffer feelings of inadequacy, anxiety, social isolation, and burnout, drawing upon the benefits of near-peer-support resources used in other educational settings. For decades, colleges have documented the mental health benefits of peer counseling and peer health education [22]. Peer-support programs are easy to implement and cost-effective—yet, it was not until the early 2000s that medical schools began to create peer -support programs to expand non-clinical wellness services beyond licensed mental health counselors [23]. Benefits of peer support have been demonstrated in a variety of health settings, from anesthesiologists [24] to nursing students [25]. Zhao et al. (2016) found peer caring and resilience improved the subjective wellbeing of both nursing and medical students [25].

Peer-support programs vary in size and structure. Generally, students are selected and trained to serve as ambassadors of wellness to provide an umbrella of non-clinical, non-judgmental, same-day support. At some institutions, peer supporters host events—such as mindfulness, sleep, and stress-reduction interventions [26]—to promote wellness and destigmatize mental health [27]. Peer-support programs can help foster a supportive culture in which students take an active role in their classmates’ wellbeing and combat feelings of competition. 

Peer-support programs help students facing a range of challenges from sub-clinical mental health struggles (i.e., test anxiety) to clerkship experiences to discrimination to relationship issues [23,28]. For example, peer support was demonstrated to reduce stigma surrounding academic stress and psychological distress [23]. Peer support is critical to encouraging students to seek support and streamlining referrals to counseling services. Students often do not feel as comfortable turning to faculty for mental health concerns [19], and therefore, peer supporters are ideally situated to recognize warning signs in a classmate. Medical students may also worry that showing signs of depressive symptoms or mental health problems may cause them to be deemed unfit by peers and professors. Peer-support programs normalize feelings of uncertainty and sadness and help students realize they are not alone.

One of the most common reasons students use peer support is academic stressors or concerns such as failed exams [28]. Further, symptoms of anxiety and depression have a higher prevalence rate in medical students when examinations are imminent, highlighting the importance for research to specify the time of year, alongside academic class, when exploring student mental wellbeing and interventions that are timed strategically near examination periods [29,30]. As such, utilization of peer support has been noted to fluctuate with common medical school milestones [23]. Peer-support services or wellness campaigns around the time of exams can reduce student suicide [31]. Moreover, they are an effective way to encourage students to reach out for help [28]. In one study, 75% of students reported that having peer support available created a supportive atmosphere even if they did not personally plan on using the program [32]. 

However, there is a gap in the literature regarding the impact that participation in these programs may have on the medical students providing the support. The extended literature on students in other educational settings suggests that students who provide peer support may also benefit [32]. In a peer program of college students who provided assistance to peers with chronic medical conditions, the providers showed high empathy, low stigma, and high confidence after providing support to their peers [33]. Similarly, in a study comparing undergraduate students trained to provide mental health peer support and student workers not trained to provide peer support, peer supporters exhibited lower avoidant coping and more belonging support, providing evidence that participation in these programs may enhance wellbeing for the supporter [34]. Medical student peer supporters have reported personal gratification in making their campuses more welcoming [23]. Peer supporters not only recommend self-care and wellness strategies to others but may end up incorporating those practices into their daily lives [35]. Being a peer supporter ultimately may allow students to develop empathy and increased self-efficacy in addressing difficult mental health topics. 

Therefore, in this study, the authors examined the impact of providing peer support to their medical student peers on several variables using a pre-post survey model of medical students at a single institution at the beginning and end of the academic year’s peer-support program. 

## 2. Materials and Methods

### 2.1. Hypotheses

The main hypotheses tested were: (1) Providing peer support to medical student peers as part of a standardized peer-support program will be associated with increased self-efficacy in identifying peers’ mental health concerns, increased empathy, and increased likelihood to assist peers with mental health problems; (2) providing peer support to medical student peers will be associated with decreased perceived mental health stigma; and (3) providing peer support will have similar effects on male and female medical students. 

### 2.2. Participants

Eligible participants (*n* = 38) were students in the peer-support program from an allopathic medical school in the southeast United States during the 2020–2021 academic year. Only medical students who were at least 18 years old and active peer supporters in good academic standing met inclusion criteria. All other medical students were excluded from participation in the study. Medical students who identified as second- to fourth-year students were given the opportunity to apply in June of 2020 to serve as a peer supporter for the academic year. There is a detailed application and review process to ensure that the student has appropriate characteristics to serve in this role (see Appendix B). 

### 2.3. Peer-Support Program

The University of Central Florida College of Medicine peer-support program consists of second- to fourth-year medical students who volunteer as peer supporters [23]. These students are taught to promote positive mental health and reduce mental health stigma through peer-to-peer interactions. The peer supporters are trained to encourage students to seek professional mental health support when indicated. All peer supporters undergo training from a licensed psychologist on privacy, active listening, QPR suicide prevention, motivational interviewing, and mindfulness/guided relaxation strategies. These individuals provide peer support through one-hour walk-in sessions and bi-annual outreach events. During the COVID-19 pandemic, the walk-in sessions were transitioned from in-person to the online video communications application, “Zoom”. 

Students receiving peer support at our institution have reported verbally to the program director that participation in peer-support sessions helped them address mental health symptoms. For example, some students who received peer support shared that they were better able to cope with feelings of sadness or stress surrounding exams. Another participant mentioned that after receiving peer support, he had less anxiety about clinical evaluations. Another shared she “felt heard” by the peer supporter in discussing a difficult break-up with her partner. First-year students have called the peer supporters “relatable”. Generally, the program is well-received and considered to create a supportive campus culture. 

### 2.4. Measures

In addition to demographics, four measures (see Appendix C) assessing empathy, self-efficacy, mental health stigma, and likelihood to assist peers with mental health problems were adapted from a program where college students assist their peers with chronic conditions [33]. Certain survey items were modified from their original wording to better address the mental health states examined in the present study. 

### 2.5. Demographics Survey 

Participants were asked to describe their gender, race, and age in years. They were also asked their year in medical school and the number of times they provided peer support during the 2020–2021 academic year. 

### 2.6. Empathy Measure

The 7-item empathy scale was adapted from the Interpersonal Reactivity Index (IRI) [36]. This questionnaire quantifies feelings of concern and sympathy for others in unfortunate circumstances. An example item on the questionnaire is: “I often have tender, concerned feelings for people less fortunate than me”. Participants answer each statement using a 5-point Likert scale with response ranges from “does not describe me at all” to “describes me very well”. Higher scores indicate higher levels of empathy. Cronbach’s alpha for this scale demonstrated good internal reliability (α = 0.817). Validity is demonstrated by the use of the IRI with medical students in other studies [37,38]. 

### 2.7. Self-Efficacy Measure

The 7-item self-efficacy scale was adapted from the Chronic Conditions Survey [33]. For each item, a scale was used from 0 = “cannot do at all” to 10 = “highly certain can do”. Example items include “confidence I can assist by using motivational skills to help students reflect on their situation”. Cronbach’s alpha for this scale demonstrated good internal reliability (α = 0.895). Support for the validity of this measure in this setting is the prior development and use in a sample of United States college students with chronic health conditions [33]. 

### 2.8. Mental Health Stigma Measure 

A 6-item measure to assess a participant’s willingness to engage with individuals with psychological conditions was used in this study. It was adapted from the Social Distance Scale used in prior studies (the only adaptation was the shift in wording from “condition” to “psychological condition”) [33,39]. Social distance scales consist of questions about participants’ willingness and comfort to engage with a given type of person. Responses to the six items used in the current study were “Definitely willing”, “Somewhat willing”, “Definitely unwilling”, or “Prefer not to answer”. Social distance scales typically have solid internal consistency and construct validity; for example, positive associations have been noted between believing that people with mental disorders are dangerous and desired social distance [40]. An example statement on the mental health stigma measure is: “How willing would you be to start a collaborative project with someone with a chronic psychological condition?” A lower willingness to engage with individuals with psychological conditions represents higher stigma towards these individuals. Cronbach’s alpha for this scale demonstrated good internal reliability (α = 0.855). 

### 2.9. Likelihood to Assist Peers with Mental Health Problems Measure

A 9-item measure to assess participants’ likelihood of assisting peers with mental health problems was adapted from the Chronic Conditions Survey [33]. Each item on this measure referenced one of nine mental health states that medical students may experience, such as “burnout”, “stress”, “loneliness”, and “imposter syndrome”. The primary question posed was: “Imagine that a new student at your medical school with the following problem needs assistance. What is the likelihood that you would volunteer to provide the assistance?” The wording for the measures regarding mental health states was developed by the researchers based on review of prior research and with input from students and a trained psychologist. For each item, students were instructed to “Please use a scale from: 0% = you are positive you would not offer to assist under any circumstances to 100% = you are positive you would offer to assist under any circumstances”. A higher percentage indicated a higher likelihood of assisting peers with mental health problems. Cronbach’s alpha for this scale demonstrated excellent internal reliability (α = 0.923). 

### 2.10. Procedure and Data Analysis

This is a within-subjects, pre-post, quasi-experimental study using a demographics survey and four self-report measures. All peer supporters (*n* = 38) were emailed the survey information twice during the 2020–2021 academic year. The survey was administered using Qualtrics software. The first survey was deployed in September 2020, shortly after the peer supporters were trained. The second survey was deployed in April 2021, after peer supporters had completed most or all of their support sessions. For each survey administration, students were given a two-week period to complete the survey. Responses were de-identified. 

IBM SPSS Statistics 26 was used to conduct descriptive and inferential statistics along with internal reliability tests. Four scales were analyzed: (1) empathy, (2) self-efficacy, (3) stigma, and (4) likelihood to assist peers with mental health problems. Each scale’s internal consistency was measured using Cronbach’s alpha. 

Pre-study versus post-study score mean rank comparisons were made using Wilcoxon signed-rank tests. Score percent for each scale were not normally distributed according to Shapiro–Wilk tests for normality; thus, a non-parametric test was chosen. To compare mean rank score differences pre-study and post-study within each gender group (e.g., comparisons within males pre- and post-study), eight additional Wilcoxon signed-rank tests were conducted. 

To compare all four scales’ pre-study and post-study scores by gender, multiple Welch’s *t*-tests were conducted given the scores were not normally distributed. An analysis was done comparing post-study survey responders versus non-responders’ baseline characteristics and scores to assess non-response bias. Categorical characteristics were analyzed using chi-square tests, while continuous characteristics and scores were analyzed using Welch’s *t*-tests.

## 3. Results

Seventy-five percent of peer supporters were in their second or third year of medical school. As most peer supporters were second- or third-year medical students, demographic information is provided for second- and third- year medical students at our institution in Table 1. 

The survey was distributed to 38 peer supporters, from which 36 completed the first survey, and 17 completed both surveys. The sample population (*N* = 17) had a median age of 25 (range: 24–27). The sample was composed of 52.9% Asian Americans and 47.1% non-Hispanic White Americans. In terms of gender, students identified as 52.9% men and 47.1% women. On average, the peer supporters who participated in this study offered seven one-hour-long peer support sessions during the 2020–2021 academic year. 

Four separate Wilcoxon signed-rank tests demonstrated that there was a statistically significant difference with moderate effect sizes between pre-study and post-study empathy scores (Z = −1.964, *p* = 0.050, r = 0.34) and self-efficacy scores (Z = −2.060, *p* = 0.039, r = 0.35) but not between pre-study and post-study stigma scores (Z = −0.142, *p* = 0.887) nor likelihood to assist peers scores (Z = −0.346, *p* = 0.730). Please see Figure 1 for a box plot of pre-study and post-study scores for each scale.

Eight Welch’s *t*-tests demonstrated a statistically significant mean baseline self-efficacy score difference between males (61.80 ± 19.01 percent) and females (80.59 ± 12.26), *t* (11.965) = −2.349, *p* = 0.037. Specifically, being a peer supporter increased males’ post-study self-efficacy and resulted in a self-efficacy score that was more comparable to the post-study self-efficacy scores of females. Further, females started with higher pre-study self-efficacy scores and demonstrated little change in self-efficacy scores between the pre-study and post-study survey period. Please see Figure 2 for a box plot of baseline self-efficacy scores comparing males and females. All other scores were similar between genders. See Table 2 for score comparisons for all four scales.

Within males, the mean scores for each scale increased, but only the increase in self-efficacy score (61.80 ±19.01 to 74.71 ± 16.81 percent) was significant (Z = −2.240, *p* = 0.025; see Table 3). Within females, mean score increased for all scales except for the self-efficacy scale, but none of the changes were statistically significant (Table 3).

No statistically significant difference in means or frequencies were observed in baseline participant characteristics or scores between post-study survey responders and non-responders (Appendix A). Non-responders are defined as peer supporters who had completed the pre-study survey but did not complete the post-study survey; therefore, their “baseline” scores are those collected at the beginning of the study. 

## 4. Discussion

There is a growing body of literature documenting the benefits of peer-support services for medical students [27,28]. With rising rates of burnout, anxiety, and depressive symptoms that occur during medical school, it is important to consider preventative wellness strategies that contribute to the protective umbrella of mental health services available to medical students. Peer-support programs present a low-cost, sustainable modality to promote student wellbeing. They do not replace clinical services, but they offer benefits in terms of engaging the medical student community in mental health advocacy and prevention, serving as a pipeline to clinical services, and creating a more supportive learning atmosphere [32]. Thus far, studies have identified the following outcomes associated with peer support for medical students and other health professionals: improving perceptions of support, reducing psychological distress, and increasing resilience [25,32].

The authors of this study sought to examine the impact that involvement in a peer-support program would have on the peer supporters themselves, specifically “how does helping affect the helpers?” This is a novel way of looking at the benefit of a peer-support program among medical students. Our findings indicate that involvement in such programs do indeed offer benefits. Specifically, the first finding of our study was that being a peer supporter increased students’ self-reported empathy and self-efficacy to help others from the start of the program over the course of a year. Role-modeling and social learning theory help explain why being a peer supporter may lead to greater empathy given that the peer supporters engage in training, monthly meetings, and interact during walk-in hours [41,42]. Being around other like-minded peers who value altruism may also contribute to why such a program promotes greater empathy [43]. In terms of self-efficacy, repetitive exposure and practice using peer support skills, such as active listening, likely increases students’ confidence and self-efficacy in applying these interpersonal skills [44,45]. 

We also found that self-efficacy seemed to increase the most among male peer supporters. Given the exploratory nature of this study, it is unclear why males had much lower baseline self-efficacy scores or showed improvement in self-efficacy. It is plausible that gender-roles may have led male supporters to have had less prior experience providing support to peers compared to their female counterparts [46,47]. Further, compared to females, even after being a peer supporter, males demonstrated lower post-study survey self-efficacy scores compared to females. This may suggest that serving as a peer supporter should complement other strategies to promote self-efficacy among men [48,49]. Women also demonstrated little change in self-efficacy scores over the course of the year, suggesting either a ceiling effect in self-efficacy as measured with this scale or perhaps that serving as a peer supporter is not an effective means to promote self-efficacy among women. Future studies with larger samples and more sensitive tools should further explore differences by gender and other aspects of identity, such as race. Two other variables examined in this study, namely mental health stigma and likelihood to volunteer to help others with mental health problems, did not yield significant changes in any of the analyses. The finding regarding no change in stigma warrants further investigation since this is a previously identified benefit of peer support although many of these studies have been theoretical or perspective pieces. More data are needed to document if peer support affects stigma over time (related to mental health or help-seeking behaviors). Still, one explanation could be that those who choose to participate as a peer supporter have less stigma toward mental health to begin with compared to the average medical student. A similar interpretation can be applied to why an effect was not observed in terms of likelihood to help others with mental health problems. 

This study has strengths and limitations that should be noted. Strengths include a novel approach to examining the benefits of providing peer support. Limitations include small sample size, lack of true randomization because peer supporters self-select to apply to the program, and lack of a control group. It is possible that medical students experience growth in terms of empathy and self-efficacy by virtue of something other than involvement as a peer supporter, such as practice of medicine skills. Thus, the limited findings should be interpreted cautiously and used as a platform for future research studying the benefits of peer support. Non-response bias can be discarded among those who failed to fill out the post-study survey, as their baseline characteristics and scores did not significantly differ from those who completed both surveys.

Despite the benefits, peer-support programs also have some barriers. A one-size-fits-all approach is not effective for all students. In particular, medical schools may not have enough students from marginalized backgrounds to address every contributor to distress, or others may not have adequate infrastructure to match students with supporters. Schools should recognize different forms of support needed for students of different identities, especially marginalized identities or international medical students. Another limitation is that students who would serve as peer supporters are sometimes themselves stressed, tired, and overworked or have compassion fatigue. Structural changes to improve the culture of medicine need to come from the top versus student-led initiatives. In addition, this highlights the importance of thorough application, screening, and training processes for selecting peer supporters to avoid any harm. Finally, medical schools are small communities, and concerns over confidentiality exist [32]. Peer-support programs need to ensure they properly respect and incorporate all aspects of identity for sexual, gender, and racial minorities in medical school.

## 5. Conclusions

Peer-support programs offer some very tangible benefits for the medical student community, and for the peer supporters themselves. This study highlights how involvement as a peer supporter increases empathy and self-efficacy. These are excellent translational skills that medical students can apply to their clinical care with patients on rotations and as they begin residency. These findings add to the growing body of literature on peer-support programs as a valuable mental health and preventative wellness resource in medical education. These findings further underscore the potential benefits not only for medical students who use the services but for those who provide them as well.

## Figures and Tables

**Figure 1 ijerph-19-05135-f001:**
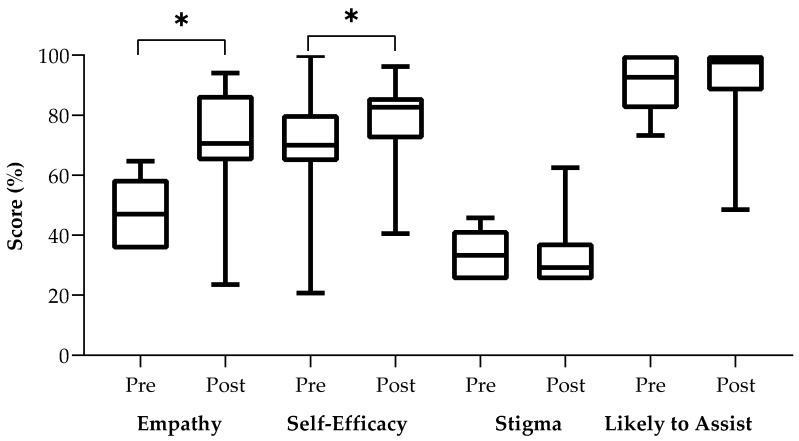
Box Plots of Pre-study and Post-study scores for Each Survey Measure. * Indicates statistically significant mean rank differences (α = 0.05) using Wilcoxon sum-rank tests.

**Figure 2 ijerph-19-05135-f002:**
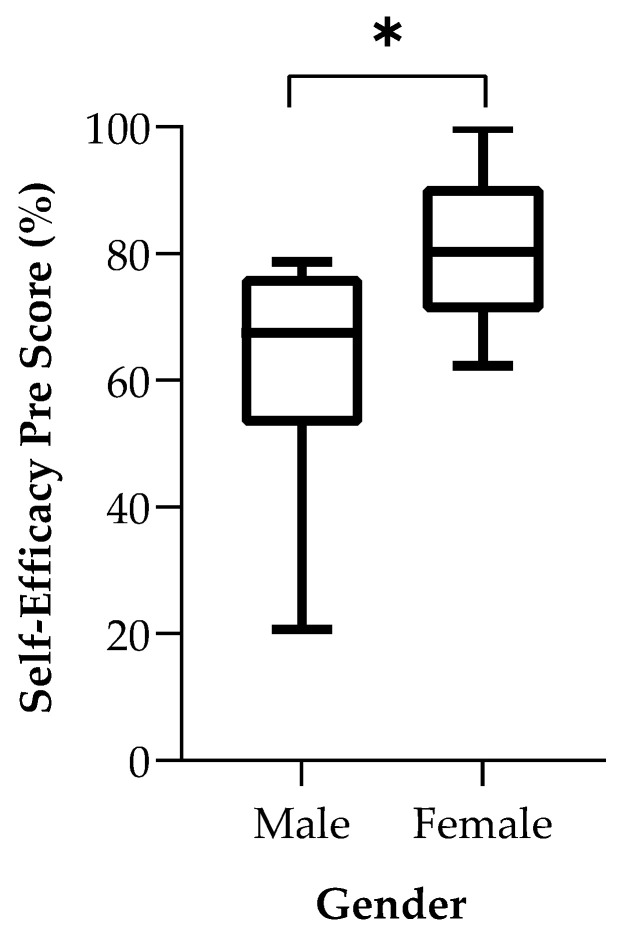
Box Plot comparing baseline Self-Efficacy Scores between males and females who completed both surveys. * Indicates statistically significant mean differences (α = 0.05) using Welch’s *t*-tests.

**Table 1 ijerph-19-05135-t001:** Demographic Information of Medical School Students.

	Second-Year Medical Students (*n* = 120)	Third-Year Medical Students (*n* = 117)
**Mean Age**	24.0 years	24.2 years
**Gender**		
Male	49% (59)	60% (70)
Female	51% (61)	40 (47)
**Race/Ethnicity**		
White/Caucasian	48% (58)	54% (64)
Asian (including Far East Asia and Pacific Islander)	32% (38)	25% (39)
Hispanic /Latino	13% (15)	11% (13)
Black/African-American	6% (7)	1% (1)
Other	1% (2)	0 (0%)

**Table 2 ijerph-19-05135-t002:** Male and Female Pre-study and Post-study Survey Score Comparisons for all Four Scales.

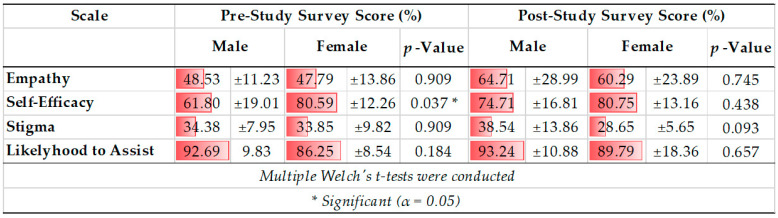

**Table 3 ijerph-19-05135-t003:** Pre- vs. Post-Study Survey Score Comparison for All 4 Scales within Males and Females.

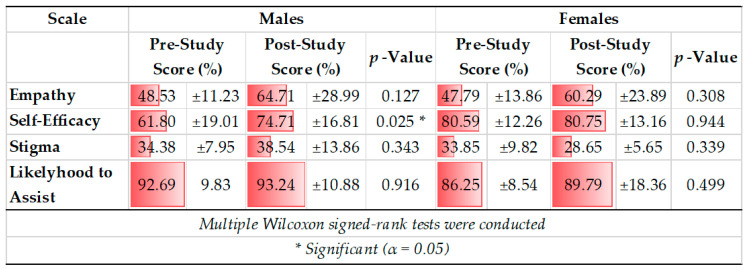

## Data Availability

Not applicable.

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
