# Peer review of "Impact of Providing Peer Support on Medical Students’ Empathy, Self-Efficacy, and Mental Health Stigma"

_ijerph, 2022, doi:10.3390/ijerph19095135_

Round 1

Reviewer 1 Report

The author present a very clearly outlined manuscript. 

METHODS/RESULTS

  • Supplemental table 1 is a bit unclear
    • The title indicates a comparison between post survey responders and non-responders at baseline. However, the table heading indicates comparison between post survey responders and post survey non responders (not at baseline)
    • Furthermore, how is data available for analysis for non responders. Non-responder would indicate that the survey response was not completed.
    • The details of this table needs to be clarified

DISCUSSION

  • The discussion lacks an exploration of plausible factors to support the findings. Possible explanations of the findings were not included
  • The discussion merely repeats the result section
  • What factors increased students' self-reported empathy and self-efficacy to help others from the start of the program over the course of a year? Was gender/race a significant factor?

Reviewer 2 Report

Given the importance of peer support described by the authors,  this particular study adds to the body of knowledge. A couple of questions and suggestions:

  1. What are the demographic characteristics of the medical student body at the medical school in question and the demographic characteristics of the entire peer mentor group in comparison to the study group?
  2. With regard to the statistical analysis were there any gender differences in the scores?
  3. Additionally, since the authors looked at differences in the scores between groups they should consider clarification of their hypothesis
  4. Finally, for the statistically significant results, what were the Wilcoxon effect sizes?
